# Change in Factors Affecting Cyberbullying of Korean Elementary School Students during the COVID-19 Pandemic

**DOI:** 10.3390/ijerph191711046

**Published:** 2022-09-03

**Authors:** Yeon-Jun Choi, So Young Shin, Julak Lee

**Affiliations:** 1Department of Aviation Security Protection, Kwangju Women’s University, Gwangju 62396, Korea; 2Department of Police Administration, Joongbu University, Geumsan 32173, Korea; 3Department of Industrial Security, Chung-Ang University, Seoul 06974, Korea

**Keywords:** cyberbullying, elementary school students, adolescents, school violence, comparative study

## Abstract

The importance of social networking and the online environment as core factors in building relationships has grown as a result of the COVID-19 pandemic, which limited in-person activities. As classes transitioned to online platforms, there was an influx of elementary school students into the cyberspace, increasing the risk of exposure to cyberbullying. This study analyzed the factors influencing the experience of cyberbullying among Korean elementary school students around 2020, when the spread of COVID-19 began in earnest, and thus suggests directions for cyberbullying prevention measures for the post-COVID-19 era. This comparative study used binary logistic regression to analyze data from the “Cyber Violence Survey” conducted by the Korea Communications Commission in 2019 and 2020. The analysis confirmed that interactions between parents and children, cyberbullying control by schools, and recognition of cyberbullying as a problem had statistically significant influences on cyberbullying experience only in 2020 (i.e., when the pandemic began). Overall, this study emphasizes the importance of raising awareness about cyberbullying among elementary school students and taking preventive action through a home-school system to address cyberbullying in the post-COVID-19 era.

## 1. Introduction

More diverse media content is developed as the information age advances, and the 4th industrial revolution, through the internet of things (IoT), artificial intelligence (AI), virtual reality (VR), augmented reality (AR), big data, and algorithms, has brought about considerable change to human lives. As the frequency of in-person activities declined following the spread of COVID-19 and relationships began forming predominantly in the virtual space through social networks and various communities, digital media have had the social side effect of increasing cyberbullying, including swearing, personal information leakage, and defamation [1]. Existing cyberbullying has been recognized as school violence, and teenagers’ subcultures have been identified in the field of youth education.

Cyberbullying is defined as the act of bullying or persistent harassment on the internet, as well as stalking, defamation, and other forms of online harassment [2,3]. Cyberbullying cannot easily be detected by a third party unless the victim makes a statement, and anyone can become a perpetrator because of the diversification of internet platforms. Additionally, perpetrators often do not recognize cyberbullying as a violent act [4], and because of the nature of cyberspace, as temporal and spatial constraints disappear, bullying and harassment in schools have expanded to cyberbullying outside of school hours [5]. Because these actions are often committed anonymously, they are difficult to detect or suppress; they are intentional and persistent, and thus attract attention as a form of school violence [6,7].

Since the lockdown from COVID-19, the public education system has switched to online classes, which leads to students spending a considerable amount of time on the internet. This social phenomenon has highlighted the severity of cyberbullying particularly among young people [8,9]; the lower the level of awareness of cyberbullying among younger age groups, the stronger the tendency to resolve conflict with friends online [10,11]. Moreover, Korea needs an alternative for active contact-free classes because of social distancing and dependence on various online platforms, which can lead to even more serious situations as elementary school students progress to higher grades. Therefore, this study analyzes the factors influencing cyberbullying among elementary school students in Korea and presents policy recommendations for cyberbullying in the post-COVID-19 era.

## 2. Theoretical Framework

School violence in Korea was on the decline until 2019, but there has been a rise in cyberbullying and its severity in other countries, especially among elementary school students [12,13,14,15]. One important characteristic of cyberbullying is that, unlike physical aggression, levels of violence tend to increase because they occur in situations where perpetrators and victims are physically separated and confrontation is not direct [16]. Because of this, children, including elementary school students, are more likely to be exposed to jokes, pranks, injuries, and privacy violations because they do not properly understand cyberbullying or take it seriously [4]. Schneider, O’Donnell, and Smith (2015) compared school violence and cyberbullying trends among more than 16,000 high school students from 2006 to 2012 [17]. While school violence decreased over time, the rate of cyberbullying increased significantly. Further, school violence, which occurred 1.7 times more in 2006, decreased by 3% in 2012; however, cyberbullying increased by 6% as a result. Particularly, many studies have shown that the rise in cyberbullying is related to the proliferation and increased use of smartphones [18,19,20,21], and the popularization of smartphones is related to increased online interpersonal relationships. In other words, decreased face-to-face interactions due to social distancing following the spread of COVID-19, active relationships on social media and in online environments, and expansion of opportunities to access various platforms may spread or create new patterns of cyberbullying. Considering these circumstances, this study analyzed changes in factors influencing cyberbullying among elementary school students during the COVID-19 pandemic in Korea.

## 3. Research Design

### 3.1. Data Collection and Measurement

This study analyzed data from the “Cyber Violence Survey” conducted by the Korea Communications Commission in 2019 and 2020 to determine the rate of cyberbullying experienced by Korean elementary school students. These data were collected through regional/grade stratified systematic sampling in Korean public schools, with a focus on the relationship between internet usage behavior and cyberbullying. The researchers recruited 1577 elementary school students consisting of 789 males and 788 females within the grades of 4 to 6 and aged between 10 and 12, living in 17 cities/provinces in Korea, who could understand and respond to the survey. In 2019, the survey was mailed to target schools randomly selected from a larger list. However, during the COVID-19 pandemic, all surveys were converted to an online format and distributed through links to students after contacting advisory teachers at the same schools chosen in 2019. The stratified sampling proceeded by organizing homogeneous subgroups from the population and extracting samples according to specific representativeness; the homogeneity of these groups helped reduce sampling errors and ensured the representativeness of related variables.

To identify the factors influencing Korean elementary school students’ experiences of cyberbullying, we first coded and cleaned the data before analyzing them. We performed frequency analysis to describe the sample’s characteristics and calculated Cronbach’s α values to assess the measured variables’ reliability. We also implemented binary logit analysis to examine correlational and causal relationships among the variables.

### 3.2. Variables

First, as shown in Table 1, the dependent variable in this research was cyberbullying experience, which includes cyber verbal abuse, cyber defamation, cyberstalking, cyber sexual violence, cyberbullying, and cyber coercion. We categorized each behavior as “experienced” or “non-experienced” (Cronbach’s α = 0.807). Moreover, we converted the dependent variable into a dummy variable with values of 0 (non-experience) and 1 (experience) for the binary logit analysis.

The independent variables for this study, selected based on prior research [22], included the following factors: reliability of friendships, number of friends perpetrating cyberbullying, frequency of exposure to harmful content, parent–child interactions, the degree to which school involvement and awareness of cyberbullying is involved in preventing or punishing cyberbullying, and experiences witnessing cyberbullying. The reliability of friendships and the parent–child interaction measure the levels of trust and interactivity between friends and family members. Exposure to harmful content measures the frequency at which students encounter content with violence, sensationalism, slander, illicit behavior, false advertising, online gambling, and other forms of online peer pressure. School involvement quantifies the degree to which schools and teachers implement programs and regulations associated with cyberbullying prevention as well as their interest in it. Awareness of cyberbullying determines the need for sanctions imposed for cyberbullying and the severity of it. Lastly, we measured the frequency of students’ witnessing of cyberbullying during the past year. Table 2 shows descriptive statistics for these variables.

## 4. Results

### 4.1. Rates of Cyberbullying before and after Onset of COVID-19

The results showed that the proportion of students reporting both perpetration and victimization of cyberbullying rose by 1.8%. By contrast, the cyberbullying perpetration rate for elementary school students dropped 0.9% from 2019 to 2020. However, the cyberbullying victimization rate increased sharply by 6.9%, as shown in Table 3.

### 4.2. Correlations between Cyberbullying Perpetration and the Variables of Interest

Table 4 presents the results of the correlational analysis. The correlation between cyberbullying perpetration and the independent variables was unclear in the 2019 data. However, the data collected in 2020 showed consistent significant relationships between cyberbullying perpetration and the independent variables.

### 4.3. Factors Affecting Elementary School Students’ Cyberbullying Perpetration Experience before and after COVID-19

To compare the factors influencing Korean elementary school students’ cyberbullying perpetration experiences before and after the COVID-19 pandemic, we included variables suggested by prior research and in a logit model, as shown in Table 5.

We can draw several conclusions from this analysis. Regardless of the COVID-19 pandemic, gender, friendship reliability, number of friends perpetrating cyberbullying, experiences witnessing cyberbullying, and frequency of exposure to harmful content all significantly influenced cyberbullying perpetration among elementary school students. In 2020, after COVID-19 became prevalent, the following factors were significant influences on cyberbullying perpetration: gender, number of friends perpetrating cyberbullying, experiences witnessing cyberbullying, frequency of exposure to harmful content, parent–child interaction, school involvement, and awareness of cyberbullying issues.

Specifically, during 2019 and 2020, the data showed that female students had 40% and 55% greater chances, respectively, of becoming cyberbullying perpetrators than their male counterparts. Furthermore, as the number of friends perpetrating cyberbullying increased, cyberbullying perpetration increased by 2.413 times in 2019 and 2.37 times in 2020. Additionally, we confirmed that 65% and 52% more students perpetrated cyberbullying when they observed cyberbullying behaviors in 2019 and 2020, respectively. Frequent exposure to harmful content also increased cyberbullying perpetration by 66% and 65% before and after the COVID-19 pandemic, respectively. Even more importantly, we observed that factors including parent–child interaction, school involvement, and awareness of cyberbullying issues did not affect cyberbullying in 2019, but affected it significantly in 2020, increasing cyberbullying perpetration by 2.519 times, 1.367 times, and 1.571 times, respectively. On the other hand, we observed that friendship reliability was the only factor influencing cyberbullying perpetration in 2019, but not 2020; with every one unit decrease in friendship reliability, cyberbullying perpetration increased by 1.611 times.

In sum, the following factors were significant influences on cyberbullying perpetration regardless of the COVID-19 pandemic: gender, number of friends perpetrating cyberbullying, witnessing cyberbullying behaviors, and exposure to harmful content. Moreover, friendship reliability was the only influential factor in 2019, and parent–child interaction, school cyberbullying control, and awareness of cyberbullying issues only influenced cyberbullying perpetration in 2020.

## 5. Discussion

The following conclusions may be drawn after comparing South Korean elementary school students’ cyberbullying perpetration experiences prior to and after the onset of COVID-19 in 2019 and 2020, respectively. First, we found that cyberbullying perpetration had little impact compared with that of victimization following the COVID-19 pandemic. On the surface, cyberbullying experience rates in 2019 and 2020 appear similar. However, recognizing the phenomenon of victims becoming cyberbullying perpetrators [23,24,25], and given that the victimization rate has also risen to an extent, we can deduce the seriousness of cyberbullying among elementary school students after the onset of COVID-19. According to Beran et al.’s (2015) research results [26], 14.0% of Canadians ages 10 to 17 have been victims of cyberbullying, and 8.0% of them have also been perpetrators of cyberbullying. Interestingly, 25.7% of students who have been victims of cyberbullying have previously cyberbullied others. Likewise, prior research on cyberbullying has consistently demonstrated that victimization and perpetration can co-occur [23,24,25]. Following the COVID-19 pandemic, especially in South Korea, the transition to online education led more elementary school students to own smartphones or laptops and to spend more time online, resulting in inadvertent exposure to cyberbullying.

Second, we discovered that gender, the number of friends perpetrating cyberbullying, and experiences witnessing cyberbullying impacted cyberbullying perpetration both before and after the onset of the pandemic. Various other studies [24,27,28,29,30,31,32,33] have implicated these factors in cyberbullying. Moreover, our study shows that these factors have been influential regardless of the circumstances created by COVID-19.

Third, friendship reliability influenced cyberbullying experience in 2019 prior to COVID-19, but not in 2020. According to research by Alarid et al. (2000) [34] and Galbavy (2003) [35], the primary cause of juvenile delinquency is a lack of trust in friendships. Nevertheless, following the COVID-19 outbreak, most schools in Korea transitioned to remote classes, resulting in relatively less interaction between classmates.

On the other hand, we discovered that the parent–child relationship, school involvement in cyberbullying, and awareness of cyberbullying issues had effects on cyberbullying experience only in 2020. Parent–child interaction has been a significant predictor of cyberbullying in prior research [36,37]. The influence of and frequent contact with parents increased as external activities decreased and time spent at home increased as a result of social distancing during the pandemic. Accordingly, elementary school students’ awareness of cyberbullying issues is important because, as online activities increased during the spread of COVID-19, students became responsible for recognizing and punishing cyberbullying themselves. Moreover, this study shows that schools’ control of cyberbullying is becoming increasingly important considering that the remote educational environment will continue to be utilized in the post-COVID-19 era. While teachers can intervene directly in physical school violence, they are often reluctant to become involved in such matters and may fail to recognize types of violence that are less obvious, including cyberbullying [38]. However, spurred by the need to teach classes in virtual space, teachers have become more aware of and interested in cyberbullying, which is likely why our results showed schools’ cyberbullying control to be influential on cyberbullying perpetration.

## 6. Conclusions

Factors affecting Korean elementary school students’ cyberbullying perpetration experience have changed as a consequence of the COVID-19 pandemic. Above all, parent–child interactions have played a significant role during the pandemic. Thus, it is crucial for parents to have routine conversations about daily life with children from a young age (especially when they are starting to be exposed to cyberbullying) to create strong bonds and allow them to monitor their children’s online activities continuously. Additionally, given the importance of cyberbullying awareness, parent–child relationships, and school intervention, it is necessary to build a strong parent–school connection to respond to cyberbullying. Recently, primary education in Korea has made many efforts to prepare for the post-COVID-19 era, such as employing new web education tools including metaverse platforms. These trends will continue in the medium-to-long term, as should preparation for active prevention of and response to cyberbullying in the post-COVID-19 era.

## Figures and Tables

**Table 1 ijerph-19-11046-t001:** Operational definitions of cyberbullying.

Type	Operational Definition
Cyber verbal abuse	The act of swearing, using harsh language, and making aggressive personal remarks via the internet, mobile phone text messages, etc.
Cyber defamation	The act of posting articles that defame other individuals/institutions on the internet or social media, regardless of whether they are true, so that an unspecified number of people can see them.
Cyberstalking	The act of sending unwanted e-mails or text messages that cause fear or anxiety repeatedly, or leaving traces of comments by visiting blogs, social media, etc.
Cyber sexual violence	Communicating sexually unpleasant content such as sexual descriptions, sexually disparaging remarks, and sexist abusive language via the internet or mobile phone, or sending obscene videos and photos to specific people.
Cyberbullying	Leaking personal information or posting personal secrets on the internet or social media, or teasing others via chatrooms, smartphones, or instant messengers.
Cyber coercion	Cyber extortion refers to the act of stealing money and smartphone data from the internet and forcing others to say/do unwanted things via internet or mobile phone.

**Table 2 ijerph-19-11046-t002:** Descriptive statistics for the variables of interest.

	2019				2020			
	Range	n	%	Mean (SD)	Range	n	%	Mean (SD)
Gender	Male	789	50.0	1.500 (0.500)	Male	876	50.4	1.496 (0.500)
	Female	788	50.0		Female	862	49.6	
Perpetration	Non-experienced	1367	86.7	0.133 (0.340)	Non-experienced	1522	87.6	0.124 (0.330)
	Experienced	210	13.3		Experienced	216	12.4	
Friendship reliability	High	1396	88.5	1.115 (0.319)	High	1256	72.3	1.723 (0.448)
	Low	181	11.5		Low	482	27.7	
Number of friends perpetrating cyberbullying	None	1431	90.7	1.107 (0.353)	None	1635	94.1	1.063 (0.260)
	1~3	123	7.8		1~3	96	5.5	
	Over 4	23	1.5		Over 4	7	0.4	
Observation	Experienced	254	16.1	1.839 (0.368)	Experienced	213	12.3	1.880 (0.328)
	Non-experienced	1323	83.9		Non-experienced	1525	87.7	
Exposure to harmful contents	High	39	2.5	2.890 (0.384)	High	268	15.4	2.283 (0.715)
	Middle	96	6.1		Middle	711	40.9	
	Low	1442	91.4		Low	759	43.7	
Parent–child interaction	High	1410	89.4	1.106 (0.308)	High	1556	89.5	1.105 (0.306)
	Low	167	10.6		Low	182	10.5	
School involvement	High	572	36.3	1.796 (0.693)	High	792	45.6	1.609 (0.606)
	Middle	754	47.8		Middle	834	48.0	
	Low	251	15.9		Low	112	6.4	
Awareness of cyberbullying	High	1506	95.5	1.045 (0.207)	High	967	55.6	1.444 (0.497)
	Low	71	4.5		Low	771	44.4	
Total		1577	100	-		1738	100	-

**Table 3 ijerph-19-11046-t003:** Changes in cyberbullying experiences from 2019 to 2020.

Year	n	Perpetration	Victimization	Both Perpetration and Victimization
2019	1577	13.3%	18.9%	7.9%
2020	1738	12.4%	25.8%	9.7%

**Table 4 ijerph-19-11046-t004:** Bivariate relationships between variables.

	Model 1 (2019)								Model 2 (2020)							
	1	2	3	4	5	6	7	8	1	2	3	4	5	6	7	8
1	1								1							
2	0.064 *	1							0.078 **	1						
3	0.235 ***	0.071 **	1						0.204 ***	0.002	1					
4	−0.240 ***	−0.015	−0.292 ***	1					−0.194 ***	−0.011	−0.301 ***	1				
5	−0.257 ***	0.015	−0.151 ***	0.206 ***	1				−0.276 ***	−0.074 **	−0.149 ***	0.143 ***	1			
6	0.035	0.225 ***	0.065 *	−0.062 *	-0.073	1			0.150 ***	0.174 ***	0.032	−0.101 ***	−0.043	1		
7	−0.017	0.060 *	−0.007	0.038	0.070 **	0.110 ***	1		0.105 ***	0.145 ***	0.066 **	−0.070 **	−0.009	0.174 ***	1	
8	0.005	0.095 ***	0.021	0.02	0.054 *	0.154 ***	0.148 ***	1	0.145 ***	0.153 ***	0.095 ***	-0.087 ***	−0.144 ***	0.152 ***	*	1

1. Cyberbullying perpetration, 2. friendship reliability, 3. number of friends perpetrating cyberbullying, 4. cyberbullying observation, 5. exposure to harmful content, 6. parent–child interaction, 7. school involvement, 8. awareness of cyberbullying issues. * *p* ≤ 0.05; ** *p* ≤ 0.01, *** *p* ≤ 0.001.

**Table 5 ijerph-19-11046-t005:** Factors affecting cyberbullying perpetration before and after the COVID-19 pandemic.

	Model 1 (2019)				Model 2 (2020)			
	B	S.E.	Wald	Odds Ratio	B	S.E.	Wald	Odds Ratio
Gender	−0.509 **	0.164	9.591	0.601	−0.791 ***	0.171	21.292	0.454
Friendship reliability	0.476 *	0.229	4.318	1.610	0.080	0.174	0.212	1.083
Number of friends perpetrating cyberbullying	0.881 ***	0.175	25.454	2.413	0.863 ***	0.234	13.605	2.370
Cyberbullying observation	−1.041 ***	0.184	32.036	0.353	−0.734 ***	0.203	13.068	0.480
Exposure to harmful content	−1.083 ***	0.158	46.702	0.339	−1.064 ***	0.115	84.975	0.345
Parent–child interaction	−0.159	0.260	0.373	0.853	0.924 ***	0.213	18.810	2.519
School involvement	−0.016	0.120	0.018	0.984	0.312 *	0.133	5.513	1.367
Awareness of cyberbullying issues	0.139	0.382	0.133	1.150	0.452 **	0.166	7.387	1.571
Constant	2.255 **	0.807	7.816	9.538	−0.648	0.736	0.776	0.523
2Log likelihood	1074.518				1059.380			
Cox and Snell’s R^2^	0.098				0.132			
Nagelkerke R^2^	0.181				0.249			
χ^2^	162.979				245.400			
Accuracy	86.7				87.6			

* *p* ≤ 0.05; ** *p* ≤ 0.01; *** *p* ≤ 0.001.

## Data Availability

The data presented in this study are openly available in “Cyber Violence Survey” conducted by the Korea Communications Commission in 2019 and 2020.

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
