# Peer review of "Change in Factors Affecting Cyberbullying of Korean Elementary School Students during the COVID-19 Pandemic"

_ijerph, 2022, doi:10.3390/ijerph191711046_

Round 1

Reviewer 1 Report

Thank you for inviting me to review this manuscript. My comments are attached.

Author Response

Reviewer 1

Comments and Suggestions for Authors

Thank you for inviting me the opportunity to review the manuscript “Change in Factors Affecting Cyberbullying of Korean Elementary School Students during the COVID19 Pandemic.” This topic is relevant to international readers as both bullying and COVID19 are worldwide epidemics that have created and continue to create ongoing challenges for schools and teachers worldwide. After reading the manuscript, I have the following recommendations for the authors:

Introduction

- The first sentence (line 27) is nonsensica

: The subject parts had been missing in editing the manuscript. We revised it as a full sentence and reflected it in the manuscript.

Consider rearranging the structure of paragraph 1 by starting with the sentence “Cyberbullying is defined as the act” (line33). Continue the paragraph through line 42. Then start a new paragraph with the information beginning in line 27. Finish by discussing the recent events in Korea (line42).

: Thank you for the recommendation. The paragraph has been rearranged as the reviewer commented. The manuscript has been revised.

- Line 36 states that “anyone can become a perpetrator due to the.” It is also true that anyone can become a victim or the same reasons.

: We agree with the reviewer’s perspective. Anyone can become a perpetrator or a victim on the situation. The reason we mentioned ‘anyone can become a perpetrator~’ in Line 36 is that we tried to explain features of perpetrators for cyberbullying.

- Specify what “recent events in Korea” (line 42) have highlighted the severity of cyberbullying

: The phase ‘recent events in Korea’ indicates the occurrence of COVID-19 and the social changes in Korea from it. So, we corrected the paragraph structure and content according to the previous comments.

- For an international audience, define the ages (or grades) of students in elementary school

: This research is focused on elementary school students grade 4 to 6, with an age ranging between 10 to 12. The information about the students was noted in Data collection and Measurement section.

Theoretical Framework

- The theoretical framework is unclear. Theoretical frameworks supporting traditional bullying (e.g., socialecological or socialinteractional theory) may not apply to cyberbullying. Maybe dynamic systems theory?

: We agree with the reviewer’s comment. The theoretical background for traditional bullying cannot be fully adopted for cyberbullying. However, for the research in cyberbullying, the relevance of the situation within the time frame and COVID-19, could be crucial variables. In addition, radical social change has occurred. Therefore, the suitability of the theoretical framework will be supplemented in our future articles. We appreciate reviewer’s kind suggestion.

- Can you cite more recent research than Scheider et al., (2015)?

Dozens of recent articles have been cited in this study. Please refer to the following.

1.   Sustainability, 13Choi, J. (2020). Effects of Family Crisis on Offline School Bullying in Elementary School Students in Korea: Mediating Effects of ADHD Symptoms, Cyber Bullying Victimization, and Anger(1), 43–57.

3.   Journal of School Social Work, 35DePaolis, K. J., & Williford, A. (2019). Pathways from cyberbullying victimization to negative health outcomes among elementary school students: A longitudinal investigation. (9), 2390–2403.

5.   Computers & Education, 176Williford, A., & DePaolis, K. J. (2019). Validation of a cyber bullying and victimization measure among elementary school aged children. (5), 557–570.

7.    Psychology in the Schools, 58Qudah, M. F. A., Albursan, I. S., Bakhiet, S. F. A., Hassan, E. M. A. H., Alfnan, A. A., Aljomaa, S. S., & AL-khadher, M. M. A. (2019). Smartphone addiction and its relationship with cyberbullying among university students. (3), 628–643.

9.   International Journal of Adolescence and Youth, 25Méndez, I., Jorquera, A. B., Esteban, C. R., & García-Fernández, J. M. (2020). Profiles of Mobile Phone Use, Cyberbullying, and Emotional Intelligence in Adolescents. (22), 9404.

11.   International journal of adolescent medicine and health, 30Palermiti, A. L., Servidio, R., Bartolo, M. G., & Costabile, A. (2017). Cyberbullying and self-esteem: An Italian study. , 136–141.

13.   The Korean Association of Police Science Review, 22Bae, S. M. (2021). The relationship between exposure to risky online content, cyber victimization, perception of cyberbullying, and cyberbullying offending in Korean adolescents. , 105946.

15.   European journal of investigation in health, psychology and education, 10Doumas, D. M., & Midgett, A. (2021). The relationship between witnessing cyberbullying and depressive symptoms and social anxiety among middle school students: is witnessing school bullying a moderator? (3), 149–160.

17.   Computers in Human Behavior, 68, 352–358.

- In addition to the reasons stated beginning in line 66, the anonymity afforded by information and communication technologies facilitates cyberbullying

Yes, the anonymity is one of the reasons for causing cyberbullying. Thank you for the constructive discussion.

Research Design

- Table 2: Align order in which variables are listed to the order that are mentioned in the text

: The order of variables has been corrected in the text.

Results

- Line 119: are the results that rose by 1.8% also from 20192020? Clarify

: Yes. That results also reflect data from 2019 to 2020.

Reviewer 2 Report

This is an important topic and overall well written. I have following minor suggestions for consideration:

1. May want to include at least some statistical outcome in the abstract.

2. Line 27 is incomplete or missing something - "As the has brought....."

3. Although listed in the tables, I suggest including overall sample size in the narrative as well and whether each table represents the exact same number of participants.

Author Response

Reviewer 2

Comments and Suggestions for Authors

This is an important topic and overall well written. I have following minor suggestions for consideration:

1. May want to include at least some statistical outcome in the abstract.

: The factors affecting the elementary school students’ cyberbullying perpetration experience are in the order of significance, followed by parent-child interactions, cyberbullying control by school, and recognition of cyberbullying as a problem. The results are summarized in the abstract.

2. Line 27 is incomplete or missing something - "As the has brought....."

: We found the missing phrase and corrected the manuscript. 

3. Although listed in the tables, I suggest including overall sample size in the narrative as well and whether each table represents the exact same number of participants.

: Thank you for kind suggestion. The sample data has been reflected in the main text.